# Effects of Soybean and Tempeh Water Extracts on Regulation of Intestinal Flora and Prevention of Colon Precancerous Lesions in Rats

Nileema R. Divate [1,†], Katharina Ardanareswari [1,2,†], Yu-Ping Yu [1], Ya-Chen Chen [1], Jiunn-Wang Liao [3] and Yun-Chin Chung [1,*]

1  Department of Food and Nutrition, Providence University, Taichung 43301, Taiwan
2  Department of Food Technology, Soegijapranata Catholic University, Semarang 50234, Indonesia
3  Graduate Institute of Veterinary Pathology, National Chung Hsing University, Taichung 40227, Taiwan
*  Correspondence: ycchun@pu.edu.tw; Tel.: +886-4-26328001-15345
†  These authors contributed equally to this work.

**Abstract:** Soybean bioactivity is significantly enhanced during tempeh fermentation. This study aimed to evaluate the efficacy of tempeh on colorectal cancer cells in vitro and colon precancerous lesions (aberrant crypt foci, ACF) in vivo. In the in vitro assay, tempeh water extract (WET) could inhibit the proliferation of Caco-2 cells. In the animal assay using 1,2-dimethylhydrazine (DMH)-induced Sprague–Dawley (SD) rats, 12-weeks daily feeding of tempeh could decrease the level of *Clostridium perfringens* in cecum contents and reduce the number of large ($\geq$4 foci) ACF in the colon of treated rats, compared to the DMH control. By the results of TOF-MS and Edman degradation, the isolated antioxidant dipeptide, tripeptides, and tetrapeptides from WET might contain methionine, proline, and lysine. The bioactive peptides in tempeh might inhibit colon cancer by suppressing the growth of *C. perfringens* in the intestinal tract.

**Keywords:** aberrant crypt foci; Caco-2 cell; colon cancer; Sprague–Dawley rats; tempeh





## 1. Introduction

In recent years, colorectal cancer (CRC) prevalence has been rising in Asian populations, with Westernized dietary choices being one of the risk factors [1]. Soybeans used to be an inseparable part of daily Asian dishes. Asians had a higher rate in converting soybean flavonoids, daidzein and genistein, into equol compared to Western populations [2]. Equol was produced by the gut microbiome during isoflavone digestion. It has been hypothesized to be responsible for multiple clinical efficacies of soybeans [2]. Soybean intake has been associated with a reduced risk of colorectal cancer in Asian women [3].

Fermentation is known to enhance the bioavailability and nutritional quality of the initial nutrients [4]. Tempeh is a popular fermented soybean-based traditional food from Indonesia. It is prepared by *Rhizophus oligosporus*, *Rhizopus oryzae*, and/or *Rhizopus stolonifera* fermentation [5], forming a compact soybean chunk bound by white mycelium. Tempeh has demonstrated various bio-functions, such as lowering serum cholesterol levels, inhibiting tumor development, improving diarrhea symptoms and iron-deficient anemia, and reducing hypertension and lipid oxidation [6]. Tempeh also improved the colon environment by enhancing the gut *Bifidobacterium* and *Lactobacillus* composition and the short chain fatty acid (SCFA) concentration [7].

Biotransformation of the soybean content during the tempeh fermentation enhanced various beneficial nutrients and boosted the antioxidant activity, mainly by the formation of various new antioxidants such as 2-hydroxyanthranilic (HAA) [8], aglycone genistein and daidzein [9], 6,7,4′-trihydroxy-isoflavone [8], and glycitein-7-O-β-D-(6-O-acetyl)–glucopyranoside [10]. In our previous study, tempeh showed greater antioxidant activities than unfermented soybeans [11].

Not only providing antioxidant activity, the phenols of the fermented product also inhibited the growth of specific microorganisms, especially Gram-negative bacteria [12]. *R. oligosporus* has been known to produce antibacterial substances during growth [13]. In addition, the genistein which is present in tempeh showed inhibitory effects on primary and metastatic colon cancer cells by alleviating oxidative stress and inflammation [14].

Only one-third of the antioxidant capacity in tempeh was contributed by isoflavones; the other two-thirds were attributed to the bioactive peptides derived from proteolyzed soy proteins during microbial fermentation [15]. The peptide lunasin from soybeans proved effective to inhibit colon cancer cell HaCT-116 by triggering apoptosis and modulating the cell cycle [16]. Considering the various bioactive compounds in tempeh, we expected the potential of tempeh in inhibiting colon carcinogenesis.

Aberrant crypt foci (ACF) are the earliest lesions in the sequence of CRC and have been used as biomarkers for CRC risks. ACF with higher crypt numbers have a higher probability to develop into tumors [17]. To mimic the formation of ACF in humans, carcinogen 1,2-dimethylhydrazine (DMH) and its derivative axozymethane (AOM) have been successfully used in animal models [18]. DMH as a site-specific DNA alkylating agent requires activation by the liver, followed by the formation of glucuronide, which conjugates to enable this pre-carcinogenic compound to enter the colon via blood circulation, and carcinogen activation by the glucuronidase-active gut microbiota [19].

The objective of this study was to evaluate the efficacy of tempeh on colorectal cancer cells in vitro and colon precancerous lesions (aberrant crypt foci, ACF) in vivo.

## 2. Materials and Methods

### 2.1. Chemicals and Media

All chemicals and solvents used were of analytical grade. Nutrition broth and Cystine Tryptic agar were purchased from Hi-Media Laboratories Limited (Mumbai, India) and BBL Microbiology Systems (Cockeysville, MD, USA), respectively. Tryptic Soy agar, Reinforced Clostridial Medium, Brain Heart Infusion broth, Lactobacilli MRS Agar, Desoxycholate Agar (DC Agar), soytone, and tryptose were purchased from Difco™ (Detroit, MI, USA). Egg yolk for typtose sulfite cycloserine (TSC) agar was purchased from local store. Dulbecco's Modified Eagle Medium (DMEM) was purchased from Gibco (Carlsbad, CA, USA). Fetal bovine serum (FBS) was from HyClone/GE Healthcare Life Sciences (Pittsburgh, PA, USA). 1.1-Diphenyl–pricryl-hydrazyl (DPPH), 3-(4,5-Dimethylthiazol-2-yl)-2,5-diphenyltetrazolium (MTT), nalidixic acid, 2,3,5-triphenyltetrazolium chloride, iodoacetic acid, polymyxin B sulfate, ferric ammonium citrate, sodium metabisulfite, D-cycloserine, and 1,2-dimethylhydrazine (DMH) were from Sigma-Aldrich (St. Louis, MO, USA). Kanamycin sulfate, yeast extract, and agar were purchased from USB (Cleveland, OH, USA). The Caco-2 cells were purchased from the Bioresource Collection and Research Centre (Food Industry Research and Development Institution, Hsinchu, Taiwan).

### 2.2. Preparation of Tempeh and Its Water Extract

Tempeh was prepared according to Chang et al. [11]. Soybeans (Glycine max var. Kaohsiung No. 10), purchased from Kaohsiung District of Agricultural Research and Extension Station (Taiwan), were washed and soaked overnight in distilled water (27 °C); after the pH was adjusted to 4.0 with lactic acid, it was cooked in boiling water for 30 min. Subsequently, it was inoculated with *Rhizopus* sp. T-3 spore suspension isolated from traditional Indonesian tempeh (approx. $10^{11}$–$10^{12}$ spores/400 g soybeans), and then fermented (35 °C, 20 h). The fresh tempeh was dried (3 h, 50 °C), ground into powder, and sterilized with $\gamma$-rays to obtain the tempeh powder. This tempeh powder was then used for the animal tests. Unfermented soybean powder was prepared with a similar method, without the inoculation and fermentation steps.

To extract the bioactive compounds, a mixture of 400 g tempeh powder and 1 L distilled water was stirred with a magnetic stirrer at 4 °C for 8 h. The mixture was centrifuged (3000× *g*, 10 min, 4 °C) and the supernatant was lyophilized, generating the water extract

of tempeh (WET) powder. WET was used for in vitro colon cancer cell inhibition tests and bio-active peptide identification.

### 2.3. Inhibitory Effect on the Proliferation of Colon Cancer Cell Line

Caco-2 cells were cultured in low glucose DMEM supplemented with 10% (*v/v*) heat-inactivated fetal bovine serum (37 °C, 5% $CO_2$). The cell viability of treated cells was determined using MTT assay [20]. The cell was seeded in $1 \times 105$ cells/well density to 96 well plate and incubated (24 h). Subsequently, the medium was removed and the WET in DMEM in particular concentration was added, followed by incubation for 24 h. After incubation, WET was replaced by DMEM-MTT (0.11 mL/well) and the plates were incubated for another 4 h. After adding dissolving solution containing 10% (*w/v*) SDS (0.1 mL/well) and incubating overnight, the absorbance was measured by an ELISA reader (Fluostar Optima, BMG Labtech, Ortenberg, Germany) at 595 nm. The inhibition rate (%) was then determined.

### 2.4. Effect of Tempeh and Soybeans on Intestinal Functions in SD Rats

2.4.1. Animals

The animal use protocol has been reviewed and approved by the Institutional Animal Care and Use Committee (IACUC Protocol No: 20051219-P07 and Approval No: 20051219-A07). Male Sprague–Dawley rats (SD) rats (6 weeks old) were purchased from BioLASCO Taiwan Co., Ltd. (Taipei, Taiwan). The initial weight of the animals is shown in the Supplementary Materials (Table S1). The rats were housed individually in stainless steel net-bottom cages with a 12/12 h light-dark cycle. Water and a standard chow diet were provided ad libitum. The temperature and humidity were kept at $25 \pm 2$ °C and $65 \pm 5\%$, respectively. The rats were weighed weekly. The rats were acclimatized for 3 days prior to the experiment. The rats were divided randomly into six groups (n = 10), i.e., control (C), DMH-induced (D), DMH-induced and supplied with 300 mg/kg BW soybeans (SL), DMH-induced and supplied with 600 mg/kg BW soybeans (SH), DMH induced and supplied with 300 mg/kg BW tempeh (TL), DMH induced and supplied with 600 mg/kg BW tempeh (TH). The soybean or tempeh powder was diluted with $dH_2O$ and given daily by gastric tube feeding during the whole 12-week study period. DMH induction (30 mg/kg BW) was performed by subcutaneous injection of 5 mg DMH in 1 mL EDTA 1 M solution, twice a week for two consecutive weeks, starting from the first day of the study period. After 12 weeks, the rats were sacrificed using $CO_2$. The blood sample was collected from the abdominal aorta, and the organs including colon, heart, liver, spleen, kidney, and cecum were harvested, weighed, and then used for further analysis.

2.4.2. Hematology Examination

Whole blood samples were taken from the abdominal aorta, and the serum was obtained from 5 mL of whole blood in the Vacutainer tubes (Franklin Lakes, NJ, USA) by centrifugation (Kubota 2010, Japan) at $775\times g$ for 10 min at 4 °C. Biochemical parameters including albumin, BUN, creatinine, glucose, protein, AST, ALT, bilirubin, cholesterol, and triglyceride were analyzed at a commercial analytical service center (Lian-Ming Co., Taichung, Taiwan).

2.4.3. Determination of Precancerous Colon Lesions

Formalin-fixed colon tissue was stained with 1% (*w/v*) methylene blue for approximately 5 min and the aberrant crypt foci (ACF) was visualized by examination of the mucosal surface under a microscope.

2.4.4. Microbiota Composition in Cecal Content

The microbiota composition in the cecal content was determined following the method of Chung et al. [21]. An amount of 1 g of cecum content was homogenized with glass beads and diluted serially with a dilution solution (0.2% gelatin, 2.5% $MgSO_4 \cdot 7H_2O$, 0.125%

$FeSO_4 \cdot 7H_2O$, 0.1% $MnSO_4 \cdot 7H_2O$, 0.125% NaCl, 0.05% L-cystein·HCl, and 0.025% resazurin solution, *w/v*). The inoculum (100 μL) was spread on Bifidobacteria iodoacetate medium–25 (BIM-25), Tryptose-sulfite-cycloserine (TSC) agar, DC and MRS agar plates in a hermetic anaerobic workstation (Electrotek 500TG workstation, Electrotek, West Yorkshire, UK) with an automated flow of a microaerophilic gas mixture (5% $CO_2$, 10% $H_2$ and 85% $N_2$) and incubated for 72 h (37 °C, RH 60%).

### 2.4.5. Evaluation of the Antioxidant Status

The harvested livers were homogenized in an ice-cold PBS buffer (pH 7.0, contained 1 mM EDTA) and centrifuged at $10,000 \times g$ at 4 °C for 15 min to collect the supernatant. The activities of the SOD, catalase (CAT), glutathione peroxidase (GPx), and glutathione reductase as well as total levels of glutathione (GSH) were measured by using assay kits from Cayman Chemical Company (Biokom, Janki, Poland). Levels of advanced oxidation protein products (AOPP) as a marker of protein oxidation and oxidative stress in the rats were determined using an OxiSelect™ AOPP Assay Kit (cat. no. STA- 318, Cell Biolabs, CA, USA). All procedures were carried out according to the manufacturer's recommendations.

### 2.5. Separation, Purification, and Identification of Antioxidant Peptides from Tempeh

An amount of WET (12 g) was dissolved in 100 mL deionized water and centrifuged at $3000 \times g$ for 10 min at 4 °C. The supernatant was filtered using an amicon ultrafilter equipped with a 5 kDa cut-off ultrafiltration membrane at 4 °C and stirred with nitrogen pressure passing through. The filtrate with a molecular weight of less than 5 kDa was collected.

The collected filtrate was loaded into a Superdex peptide HR 10/30 column (1.0 × 30 cm), using water as the mobile phase and a flow rate of 0.25 mL/min for separation. Each 1 mL eluate was collected by automatic collector and aliquots with DPPH scavenging activity were subjected for lyophilization. The DPPH scavenging activity was determined according to [11].

The resulting lyophilized powder was dissolved with 25 mM Bis-Tris-HCl buffer solution (pH 7.1) and then loaded into a Mono P column (0.5 × 20 cm). The separation was carried out by eluting with 25 mM Bis-Tris-HCl buffer solution (pH 7.1) at a flow rate of 0.25 mL/min and each 1 mL eluate was collected in a tube. After collecting 30 tubes, the mobile phase was changed to polybuffer 74 (pH 3.0) and the elution was continued to 40 tubes more. Each collected fraction was tested for DPPH scavenging activity, and the fraction with DPPH scavenging activity was lyophilized.

The lyophilizate collected from the MonP chromatography was dissolved with distilled water, and 10 μL of liquid was injected into a preparative C18 column (Vercopak liquid chromatography column, 1 × 25 cm). Linear gradient chromatography was used with a mobile phase of A (0.1% (*v/v*) TFA aqueous solution) and B (acetonitrile) at a flow rate of 1 mL/min (0–55% B, 55 min). Each compound emitting main absorption peaks under a 280 nm wavelength was collected and lyophilized.

Time-Of-Flight Mass Spectrometry (TOF-MS) and Edman degradation were applied to analyze the peptide molecular weight and the amino acid sequence of the lyophilized powder with DPPH scavenging activity. The N-terminal sequencing of protein was performed following the Edman degradation method. A total of four cycles was performed, and Dptu and tup were used as reaction reagents.

### 2.6. Statistical Analysis

The data from all experiments were evaluated by one-way ANOVA, followed by Duncan's new multiple range test to determine the differences among means using the Statistical Analysis System (SAS Institute, Cary, NC, USA). A significance level of 95% was adopted for all comparisons.

## 3. Results

### 3.1. Antioxidative Oligopeptides in WET

There were three groups of signal values, and they were 231 *m/z* and 253 *m/z*, 346 *m/z* and 368 *m/z*, and 461 *m/z* and 483 *m/z*. The distance between the three groups was 115 *m/z*, and the distance between each group was 22 *m/z* (Figure 1). It was presumed that the analytes were a double peptide containing two amino acids, a tripeptide containing three amino acids, and a tetrapeptide containing four amino acids. It was inferred from the distance of 115 *m/z* that the tripeptides and tetrapeptides may contain asparagine (molecular weight 114.11 Da).

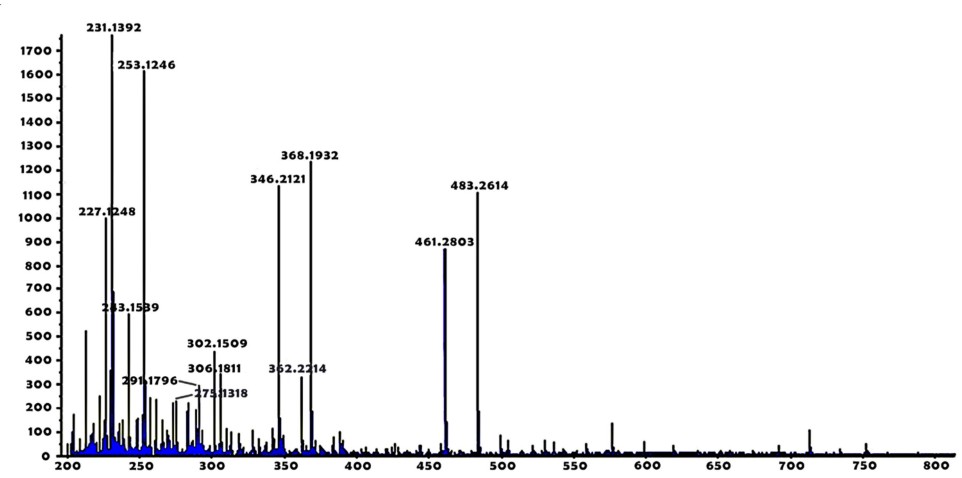

**Figure 1.** MALDI-TOF MS spectrum of antioxidant peptides isolated from the water extract of tempeh.

The isolated antioxidant peptides were likely to contain methionine, proline and lysine (Figure 2), however the attempt to sequence the amino acids failed due to the presence of many dipeptides, tripeptides, and tetrapeptides in the water extract, rather than a single peptide.

### 3.2. In Vitro Experiments

Inhibitory Effect of WET on the Proliferation of Colon Cancer Cell Line

The cytotoxic effect of WET to Caco-2 cells was exhibited in a dose-dependent manner (Figure 3). At the concentration of 200 µg/mL, the inhibitory effect of WET on Caco-2 cells was about 29%, and when the concentration of WET increased to 600 µg/mL, the inhibitory effect increased to 38.05%.

### 3.3. In Vivo Experiments

3.3.1. General Physiological and Biochemical Status of Experimental Rats

In the animal study, there was no significant difference in the body weight, food intake, and relative organ weights (heart, liver, spleen, kidney) among groups after 12 weeks of soybean and tempeh administration ($p > 0.05$, Supplemental Materials Tables S1 and S2). The histopathological study showed no significant lesions in the organs of all animals (Supplementary Materials Figures S1–S4). The serum glucose and triglyceride concentrations in the DMH-induced group was significantly lower than that of the normal control group ($p < 0.05$, Table 1). The triglyceride level in both the SL and SH groups were recovered by the soybean treatment, but the glucose level was not affected. Although the bilirubin level in the DMH-induced group was the highest, the bilirubin concentration in all groups was within normal range.

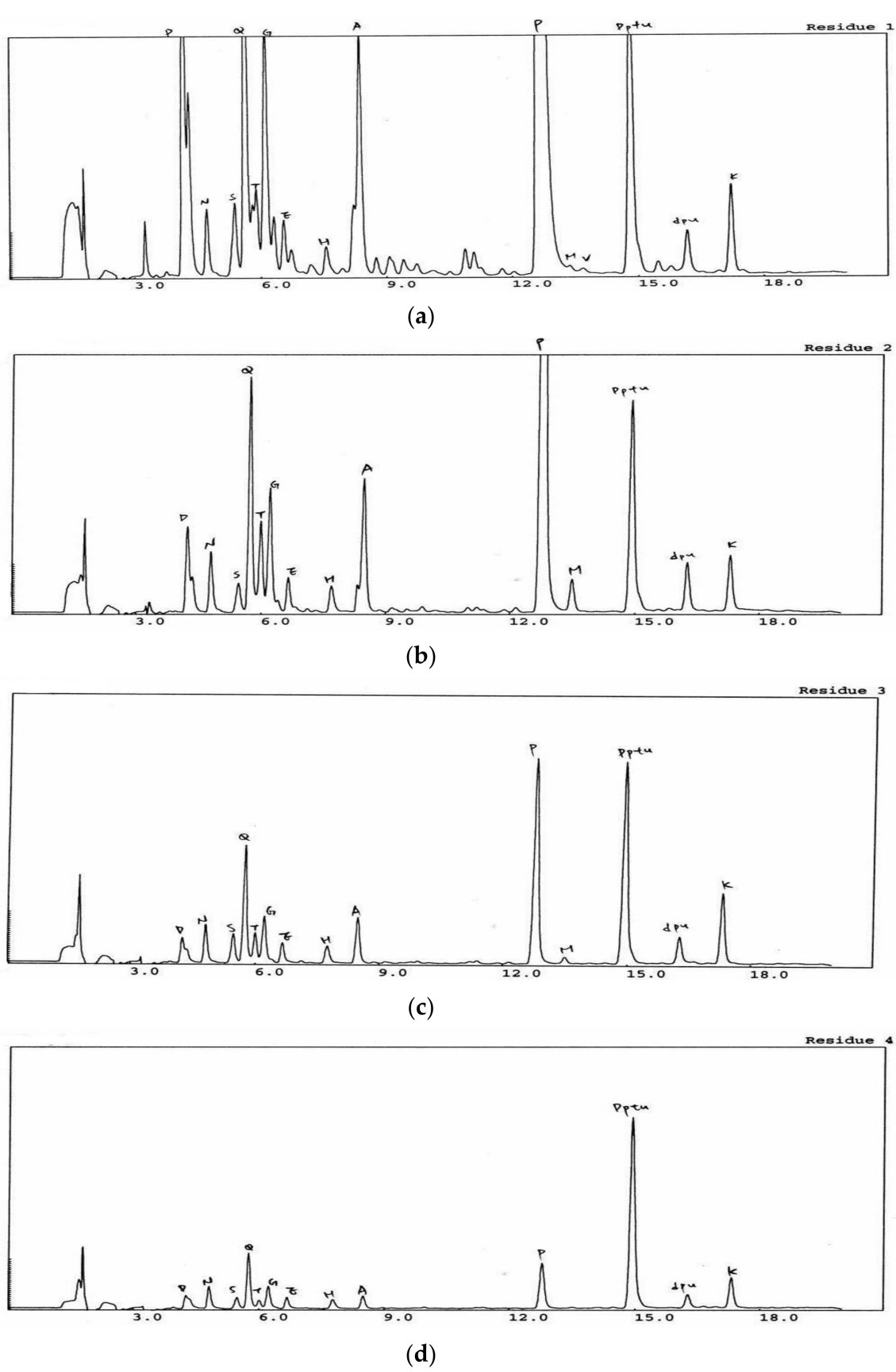

**Figure 2.** N-terminal sequencing of antioxidant peptides isolated from the water extract of tempeh from Residue 1 (**a**), Residue 2 (**b**), Residue 3 (**c**), and Residue 4 (**d**).

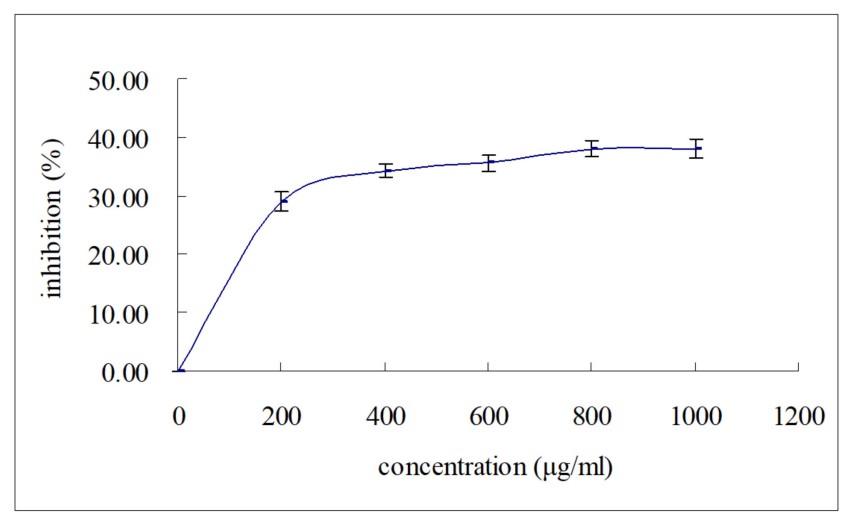

**Figure 3.** Effect of water extract of tempeh on the proliferation of Caco-2 cells.

**Table 1.** Serum biochemical parameter in rats of SD rat gavaged with soybeans or tempeh for 12 weeks.

|  | C | D | SL | SH | TL | TH |
|---|---|---|---|---|---|---|
| Albumin (g/dL) | 4.48 ± 0.28 [b] | 4.35 ± 0.30 [ab] | 4.21 ± 0.36 [a] | 4.36 ± 0.22 [ab] | 4.28 ± 0.13 [ab] | 4.33 ± 0.19 [ab] |
| BUN (mg/dL) | 13.20 ± 1.55 | 11.70 ± 2.00 | 13.33 ± 1.94 | 12.50 ± 1.08 | 11.78 ± 2.28 | 12.56 ± 1.42 |
| Creatinine (mg/mL) | 0.44 ± 0.06 | 0.44 ± 0.04 | 0.44 ± 0.08 | 0.42 ± 0.05 | 0.40 ± 0.05 | 0.41 ± 0.08 |
| Glucose (mg/dL) | 290.80 ± 65.76 [b] | 206.40 ± 50.94 [a] | 232.44 ± 75.08 [a] | 224.40 ± 44.81 [a] | 188.78 ± 35.95 [a] | 196.00 ± 41.17 [a] |
| Protein, total (g/dL) | 6.02 ± 0.40 | 5.90 ± 0.34 | 5.70 ± 0.49 | 5.89 ± 0.40 | 5.81 ± 0.29 | 5.86 ± 0.35 |
| AST (U/L) | 95.91 ± 59.21 | 121.10 ± 45.25 | 143.78 ± 89.16 | 149.10 ± 105.27 | 140.44 ± 80.40 | 134.88 ± 59.13 |
| ALT (U/L) | 50.3 ± 32.64 | 47.90 ± 18.17 | 57.67 ± 21.51 | 73.60 ± 52.22 | 67.22 ± 38.82 | 76.11 ± 55.25 |
| Bilirubin-Total (mg/dL) | 0.42 ± 0.15 [a] | 0.63 ± 0.28 [b] | 0.55 ± 0.17 [ab] | 0.52 ± 0.18 [ab] | 0.54 ± 0.13 [ab] | 0.52 ± 0.12 [ab] |
| Cholesterol (mg/dL) | 43.9 ± 12.91 [b] | 33.30 ± 11.18 [ab] | 35.22 ± 10.52 [ab] | 34.00 ± 5.62 [ab] | 28.33 ± 5.41 [a] | 34.56 ± 13.79 [ab] |
| Triglyceride (mg/dL) | 160.00 ± 93.66 [b] | 99.20 ± 34.38 [a] | 115.11 ± 47.39 [ab] | 113.00 ± 33.79 [ab] | 102.67 ± 24.52 [a] | 101.33 ± 38.34 [a] |

C: control, D: DMH, SL: DMH + 300 mg/kg BW soybeans, SH: DMH + 600 mg/kg BW soybeans, TL: DMH + 300 mg/kg BW tempeh, TH: DMH + 600 mg/kg BW tempeh; values were means ± S.D. For C, D, SH groups, n = 10; SL, TL, TH groups is n = 9; [ab] different letters in the same row indicate a significant difference as determined by ANOVA and Duncan's test; BUN, blood urea nitrogen; AST, aspartate aminotransferase; ALT, alanine aminotransferase.

The DMH induction did not produce obvious tumors, but the colon weight in the D group increased compared to the control (C) group (Table 2), which might be caused by the occurrence of ACF (Figure 4).

**Table 2.** Comparisons of colon weight, cecal weight, cecal content, pH, and fecal moisture among groups.

| Group | C | D | SL | SH | TL | TH |
|---|---|---|---|---|---|---|
| colon weight (g) | 0.09 ± 0.03 [a] | 0.11 ± 0.02 [ab] | 0.11 ± 0.03 [ab] | 0.11 ± 0.02 [ab] | 0.11 ± 0.02 [ab] | 0.12 ± 0.03 [b] |
| cecum weight (g) | 3.83 ± 1.00 | 4.04 ± 1.55 | 4.26 ± 1.14 | 3.86 ± 0.72 | 4.03 ± 0.51 | 4.00 ± 0.87 |
| cecum content (g) | 2.77 ± 0.86 | 2.75 ± 1.51 | 2.54 ± 1.38 | 2.76 ± 0.66 | 2.41 ± 0.98 | 2.36 ± 1.15 |
| pH | 6.83 ± 0.08 | 6.74 ± 0.15 | 6.88 ± 0.10 | 6.83 ± 0.09 | 6.89 ± 0.12 | 6.78 ± 0.19 |
| fecal moisture (%) | 82.53 ± 4.98 [b] | 80.40 ± 1.38 [ab] | 78.04 ± 3.73 [a] | 79.42 ± 1.26 [ab] | 78.38 ± 3.67 [a] | 80.72 ± 2.96 [ab] |

C: control, D: DMH, SL: DMH + 300 mg/kg BW soybeans, SH: DMH + 600 mg/kg BW soybeans, TL: DMH + 300 mg/kg BW tempeh, TH: DMH + 600 mg/kg BW tempeh; values were means ± S.D. For C, D, SH groups, n = 10; SL, TL, TH groups is n = 9; [ab] different letters in the same row indicate a significant difference as determined by ANOVA and Duncan's test.

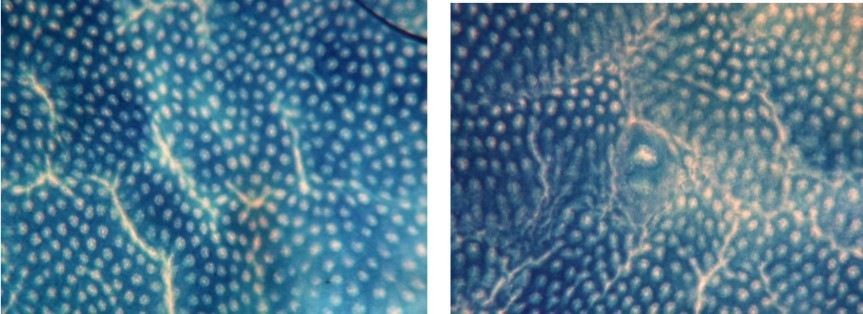

**Figure 4.** Methylene blue stain of the distal colon (**left**) normal crypt, (**right**) aberrant crypt foci (magnification 100×).

### 3.3.2. Aberrant Crypt Foci Formation

The ACF was formed in the DMH-induced group, indicated by a thick pericryptal area (Figure 4). Compared to the other groups, the normal control group had fewer ACFs (Table 3). The number of large ACF and total ACFs were significantly higher in the DMH-induced group than the normal groups ($p < 0.05$).

**Table 3.** Effect of soybeans or tempeh on DMH-induced aberrant crypt foci formation in rat colon.

| Number of Crypt/ACF | 1 Crypt | 2 Crypt | 3 Crypt | $\geqq$4 Crypt | Total ACF |
|---|---|---|---|---|---|
| C | 0.1 ± 0.3 [a] | 0.0 ± 0.0 [a] | 0.0 ± 0.0 [a] | 0.0 ± 0.0 [a] | 0.1 ± 0.3 [a] |
| D | 29.8 ± 13.4 [c] | 43.5 ± 19.4 [c] | 28.1 ± 15.4 [b] | 23.2 ± 17.2 [c] | 124.6 ± 54.9 [c] |
| SL | 11.2 ± 6.9 [ab] | 20.8 ± 13.9 [b] | 14.6 ± 11.9 [b] | 12.3 ± 12.9 [bc] | 58.9 ± 41.2 [b] |
| SH | 20.2 ± 19.4 [bc] | 33.0 ± 31.5 [bc] | 25.3 ± 28.9 [b] | 11.4 ± 12.8 [abc] | 89.9 ± 91.1 [bc] |
| TL | 15.9 ± 8.1 [b] | 28.6 ± 10.8 [bc] | 15.6 ± 6.1 [b] | 15.3 ± 8.9 [bc] | 75.3 ± 25.2 [bc] |
| TH | 17.6 ± 15.1 [b] | 27.2 ± 28.9 [bc] | 15.0 ± 11.7 [b] | 8.6 ± 13.0 [ab] | 68.3 ± 56.8 [b] |

C: control, D: DMH, SL: DMH + 300 mg/kg BW soybeans, SH: DMH + 600 mg/kg BW soybeans, TL: DMH + 300 mg/kg BW tempeh, TH: DMH + 600 mg/kg BW tempeh; values were means ± S.D. For C, D, SH groups, n = 10; SL, TL, TH groups is n = 9; [abc] different letters in the same column indicate a significant difference as determined by ANOVA and Duncan's test.

Although either the SL group or SH did not significantly lower the number of large ACF containing $\geqq$4 crypts compared with the DMH-induced group, they could reduce the total ACFs by 53.0% and 27.8%, respectively (Table 3). A lower soybean dose was more effective on reducing the total ACF in this study compared with the high dosage.

Compared with the DMH group, the administration of 600 mg/kg BW tempeh could significantly reduce the total number of ACF, and there was a dose response manner, especially on the large ACF ($\geq$4 crypts) number. After further calculating the inhibition effect (%), 600 mg/kg BW and 300 mg/kg BW tempeh could reduce the large ACF number up to 62.9% and 34.1%, respectively, compared with the DMH group (Table 3).

A low dose of soybeans (300 mg/kg BW) had the best effect on reducing the total number of ACF, even better than tempeh. However, 600 mg/kg BW tempeh was the most effective in reducing the large ACF number. Since the probability of ACF developing into a colon cancer tumor is directly proportional to the number of crypts it contains, tempeh may have the greatest potential against colon cancer.

Overall, although feeding 300 mg/kg BW soybeans had the best effect on reducing the total number of ACF, the 600 mg/kg BW tempeh treatment was the most effective among the four treatments in terms of its ability to reduce the number of ACFs containing more than 4 crypts. Since the probability of ACF developing into a colon cancer tumor is directly proportional to the number of crypts it contains, tempeh may have the greatest potential against colon cancer.

### 3.4. Microbiota Populations in the Cecum Contents

Either DMH or tempeh treatment did not cause a change in the level of *E. coli*, *Lactobacillus* spp., or *Bifidobacteria* spp. (Table 4, $p > 0.05$). The pathogenic *C. perfringens* appeared in the DMH-induced group but was suppressed in the soybean- and tempeh-treated groups (SL, SH, TL, TH).

**Table 4.** The comparison of cecal microbiota (log CFU/g dry content) among groups.

| | C | D | SL | SH | TL | TH |
|---|---|---|---|---|---|---|
| *E. coli* | 9.74 ± 0.94 | 9.95 ± 1.20 | 9.40 ± 1.13 | 9.70 ± 0.97 | 9.82 ± 0.86 | 10.1 ± 1.18 |
| *Lactobacillus* spp. | 11.78 ± 1.00 | 11.81 ± 0.74 | 11.82 ± 0.85 | 11.88 ± 0.78 | 11.70 ± 0.72 | 11.73 ± 0.87 |
| *Bifidobacteria* spp. | 8.43 ± 0.22 | 8.41 ± 0.33 | 8.19 ± 0.70 | 8.20 ± 0.50 | 7.91 ± 0.29 | 8.06 ± 0.25 |
| *C. perfringens* | ND [#] | 2.25 ± 0.52 | ND | ND | ND | ND |

C: control, D: DMH, SL: DMH + 300 mg/kg BW soybeans, SH: DMH + 600 mg/kg BW soybeans, TL: DMH + 300 mg/kg BW tempeh, TH: DMH + 600 mg/kg BW tempeh; values were means ± S.D. For C, D, SH groups, n = 10; SL, TL, TH groups is n = 9. Three replicates were performed for each sample; ND [#]: No colony was detected when 0.1 mL (0.1 g/mL) of sample was spread on the TSC agar. There was no significant difference among groups for *E.coli*, *Lactobacillus* spp., *Bifidobacteria* spp.

### 3.5. Antioxidant Effect of Soybeans and Tempeh on Liver of SD Rats

The SOD activity in the liver of the SL group was significantly increased, but the TL was decreased compared to the D group ($p < 0.05$, Figure 5), but the GLU, AOPP, GR, CAT, and GPx were not affected by either soybean or tempeh treatments.

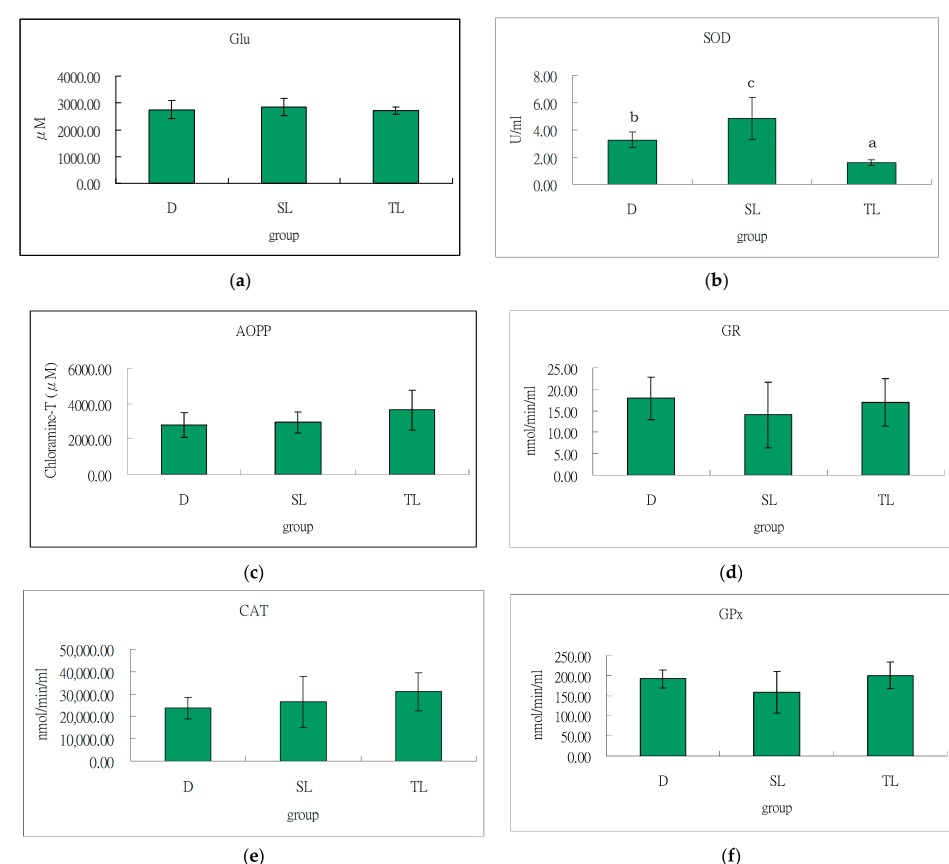

**Figure 5.** Effect of soybeans and tempeh on liver (**a**) glutathione (Glu), (**b**) superoxide Dismutase (SOD), (**c**) advanced oxidation protein products (AOPP), (**d**) glutathione reductase (GR), (**e**) catalase (CAT), (**f**) glutathione peroxidase (GPx) of SD rats. D: DMH, SL: DMH + 300 mg/kg BW soybeans, TL: DMH + 300 mg/kg BW tempeh. For D group, n = 10; SL, TL groups is n = 9; [abc] different letters indicate a significant difference as determined by ANOVA and Duncan's test.

## 4. Discussion

The inhibition of WET in the in vitro Caco-2 cells was translated in the in vivo DMH-induced rats. The aberrant crypt foci (ACF) are the earliest neoplastic lesions in the colon [22]. ACF has a thicker pericryptal area, a slightly elevated appearance above the surrounding mucosa, and a higher staining intensity [17]. We demonstrated that soybeans and tempeh could inhibit the formation or progression of ACFs, as seen in the decreasing total ACF and large ACF number.

While the pathogenic strains of *C. perfringens* were inhibited in the in vivo studies, the probiotic strains were not increased by soybean or tempeh treatment. *C. perfringens* and its enterotoxin showed high association with colorectal cancer [23]. *C. perfringens* is a major β-glucuronidase producer in the gut [24]. Some colon-specific pre-carcinogens need to be bound to glucuronic acid while being metabolized in the liver and transported to the gut, where the carcinogens are released by the β-glucuronidase-active microbiota. [25]. Higher β-glucuronidase activity has been reported in CRC patients [26]. Inhibition of gut microbial β-glucuronidase activity could prevent tumorigenesis [27]. The lower the *C. perfringens* level in the gut, the lower the level of β-glucuronidase activity and the less risks of ACF development [28].

Isoflavones in tempeh were reported to be able to elevate CAT, SOD, GR, and GSH levels in the rat brain [29]; however, the present study did not determine antioxidant activity in the rat brain, but in the liver. Our results showed that tempeh only increased the hepatic SOD. SOD's role is crucial in human antioxidant defense mechanisms by converting highly reactive superoxide anion radicals to less reactive $H_2O_2$. Increasing antioxidative enzyme activity on DMH-animal models could be interpreted as increasing protection towards lipid peroxidation and oxidative stress during tumorigenesis [30], which probably could explain the efficacy of tempeh in decreasing some ACF parameters.

During the soybean fermentation, *R. oligosporus* decomposes daidzin and genistin in soybeans into anti-cancer daidzein and genistein [31]; thus, tempeh contains doubled genistein and daidzein content compared to unfermented soybeans [29,32]. However, only one third of the antioxidant capacity in tempeh was contributed by isoflavones; the other two thirds were attributed to the peptides derived from hydrolyzed soy proteins during microbial fermentation [15]. In the previous study, a 3.5 kDa antioxidant peptide was isolated from tempeh. The antioxidative properties were speculated to belong to its special amino acid composition or structure [15]. In our study, the molecular weight of the isolated antioxidant oligopeptides was 200~500 Da and might contain methionine, proline, and lysine; but histidine was not detected. The dissolved antioxidative peptides of tempeh may exert their anti-colorectal cancer ability by alleviating the increase in the basal reactive oxygen species (ROS) level of the aberrant crypts, thus preventing the progression of colon tumorigenesis [33].

## 5. Conclusions

The potential of tempeh in suppressing colon carcinogenesis was elaborated in this study. In vitro cell culture experiments proved that WET had anti-proliferative effects on Caco-2 cancer cell lines. In the in vivo experiment, tempeh exerted its anti-colorectal cancer ability by preventing the onset of ACF and retarding the progression of ACF in DMH-induced rats. The mechanisms could be attributed to the antioxidative dipeptides, tripeptides, and tetrapeptides that have been successfully isolated in this study, which might cause the elevation of antioxidant enzyme activity in the liver. In addition, tempeh also probably prevents activation of potential carcinogens in the gut by suppressing the β-glucuronidase activity producer, *Clostridium perfringens*.

**Supplementary Materials:** The following supporting information can be downloaded at: https://www.mdpi.com/article/10.3390/pr11010257/s1, Table S1: Changes of body weight and food intake in Sprague-Dawley rat gavaged with soybean or tempeh for 12 weeks; Table S2: Relative organ weights of SD rat gavaged with soybean and tempeh for 12 weeks; Figure S1. No significant histopathological changes of control rats in the oral toxicity test; Figure S2. No significant histopathological changes of rats in the DMH group; Figure S3. No significant histopathological changes of rats in the SH group; Figure S4. No significant histopathological changes of rats in the TH group.

**Author Contributions:** Methodology, Y.-P.Y. and Y.-C.C. (Ya-Chen Chen); formal analysis, N.R.D.; investigation, N.R.D.; data curation, K.A., N.R.D. and J.-W.L.; writing—original draft preparation, N.R.D.; writing—review and editing, Y.-C.C. (Yun-Chin Chung), K.A. and J.-W.L.; supervision, Y.-C.C. (Yun-Chin Chung) and J.-W.L.; project administration, Y.-C.C. (Yun-Chin Chung). All authors have read and agreed to the published version of the manuscript.

**Funding:** This research was supported by the Ministry of Science and Technology, R.O.C. Taiwan (MOST 110-2320-B-126-003-MY3). Its financial support is greatly appreciated.

**Data Availability Statement:** The data presented in this study are available on request from the corresponding author.

**Conflicts of Interest:** The authors declared no potential conflicts of interest with respect to the research, authorship, and/or publication of this article.

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
