# Peer review of "Effects of Soybean and Tempeh Water Extracts on Regulation of Intestinal Flora and Prevention of Colon Precancerous Lesions in Rats"

_processes, doi:10.3390/pr11010257_

Round 1

Reviewer 1 Report

Comments for the authors

Major comments

Materials and Methods

-       Please provide more details about the appropriate ethical approval of this trial (approval number)

-       Please provide more details about the selected rats: breed, mean BW, diet …

-       L121-128: add appropriate references 

Results 

-       Table 1: add values per parameter

-       L184: ‘’The histopathological study showed no significant lesions in organs of all animals (Data not shown).’’ You should remove the sentence or provide more details about histopathological results.

Discussion 

-       You should discuss further the results of your study. Focus more on the significance of your results for anti-colon cancer activities

Minor comments

§  Table 2: … faecal .. / add value ‘’g’’ in colon weight 

Reviewer 2 Report

In the manuscript, the author presented a study of the effects of water extracts of soybean and tempeh (fermented soybean) on intestinal flora and colon lesions in rats. The idea of the study is acceptable, but the writing needs to be improved. Several points need to be modified and revised to improve the manuscript, as follows:

1.      In the introduction section:

·   The abbreviations CRC, ACF, and AOPP could not be used for the first time without full names; please revise the similar abbreviations in the manuscript.

·   References 15 and 16 should be updated to include the previous sentences.

2.      In the Materials and Methods section:

·   The animal approval number is missing.

·   DMH’s full name is missing, and the compound is not listed in the chemicals part. What it induced was missing. Also, no information about its induction role was found in the introduction section.

·   Did the author administer the WES or WET during the DMH induction period or after? It is not clear.

·   Why did the author use WES- or WET-specific doses?

3.      In the results section:

·   The inhibitory effect of WET seems to be small as if the IC50 were calculated, it would be over 1000 μg/ml, and this is not considered an anticancer agent.

·   Why did the author study the inhibitory effect of WET without comparing it with WES? WES effect against Caco-2 cells must be added. 

·   The author stated, "WES, but not WET, recovered the triglyceride level, but glucose level was not affected.", and the groups were SL, SH, TL, and TH. The author must give the group only one name, as it is confusing to the reader to guess whether it is WET in a low dose or in a high dose.

·   Biochemical parameters in serum were missing from the chemicals and methodology sections.

·   In table 1, the SD of several parameters is high, even higher than the mean in some results! The author should revise these results.

·   The author stated, "The DMH induction did not produce obvious tumors, but the weight of the colon and cecum walls was increased (Table 2)." increased compared to what? Also, in table 2, except for the control, the colon weight in all groups was increased, What explanation could the author provide?

·   Cecum weight (g) increased in both the treated and DMH groups. The author could give an explanation.

·   "Oral admission of high dose WES (600 mg/kg BW) could prevent the thickening of the cecum wall, but WET did not exhibit such effect", How could the author claim that WET reduced the number of ACF in the colon of treated rats while not reducing cecum wall thickness?

·   In Figure 2 the results of the treated groups are missing. The author should add the histopathological results of the ACF in all treated groups to confirm their results.

·   The author claimed that "the higher dosage (600 mg/kg BW) of WES was more effective in reducing the ACF compared with the lower dosage (300 mg/kg BW)." However, in table 3 the results are contradictory.

·   "The large ACFs in both the TL and TH groups were significantly reduced by 34.1 % and 62.9 %, respectively, as well as on the total ACF (p<0.05)." reduced compared to what group? The author should revise all the results for clear understanding.

·   "The inhibition effect of the high dose WET (600 mg/kg BW) was twice as compared to the low dose (300 mg/kg BW) (Table 3)." What is the inhibitory effect? It is not understandable.

·   How does the author calculate the inhibition effect from the total ACF? The author must make the comparisons clear to the readers.

·   The * in the table legend "Values were ....." should be removed.

·   In "3.4. Antioxidant Effect of WES and WET on the Liver of SD Rats," the SH and TH groups were missing. Also, the glutathione levels are missing in the text. The author must revise the figures and the texts.

·   Some figures’ resolution is poor.

·   "5. The oligopeptides in the WET" section must be transformed before the in vitro study.

4.      In the discussion section:

· The author stated in the methodology section that "The organs including heart, liver, spleen, kidney, and cecum, were harvested, weighed, and then used for further analysis." And in the results section, they only measure the antioxidant in the liver; however, in the discussion section, the author claimed, "Isoflavones in tempeh was reported to be able to elevate CAT, SOD, GR and GSH level in rat brain [25], but in our results the WET only increased the liver SOD." Did the author measure the antioxidants in all organs and compare them with the liver? The author must explain.

5.      The conclusion section must be rewritten to prove the idea of the study.

Reviewer 3 Report

State whether the solutions are made in the w/v or v/v relation;

please review the sentence "A significance level of 5 % was adopted for all comparisons.". Wouldn't that be 95%?

the analyzes are little discussed, please significantly improve the discussion of the manuscript.

Overall the manuscript has great potential, after improvements I will be happy to recommend it for publication.

Round 2

Reviewer 2 Report

I think the authors have addressed all the needed comments.

Reviewer 3 Report

After the review step I recommend the manuscript "Effects of soybean and tempeh water extracts on regulation of intestinal flora and prevention of colon precancerous lesions in rats" for publication.